

# Evaluation of snow cover properties in ERA5 and ERA5-Land with several satellite-based datasets in the Northern Hemisphere in spring 1982-2018

Kerttu Kouki[1], Kari Luojus[1], Aku Riihelä[1]

[1]Finnish Meteorological Institute, Helsinki, P.O. Box 503, 00101, Finland

*Correspondence to*: Kerttu Kouki (kerttu.kouki@fmi.fi)

**Abstract.** Seasonal snow cover of the Northern Hemisphere (NH) greatly influences surface energy balance, hydrological cycle, and many human activities, such as tourism and agriculture. Monitoring snow cover at continental scale is only possible from satellites or using reanalysis data. The aim of this study is to analyze timeseries of surface albedo, snow water

equivalent (SWE), and snow cover extent (SCE) in spring in ERA5 and ERA5-Land reanalysis data and to compare the timeseries with several satellite-based datasets. As satellite data for the SWE intercomparison, we use bias-corrected SnowCCI v1 data for non-mountainous regions and the mean of Brown, MERRA-2 and Crocus v7 datasets for the mountainous regions. For surface albedo, we use the black-sky albedo datasets CLARA-A2 SAL, based on AVHRR data, and MCD43D51 based on MODIS data. Additionally, we use Rutgers and JAXA JASMES SCE products. Our study covers

land areas north of 40 °N and the period between 1982 and 2018 (spring season from March to May). The analysis shows that both ERA5 and ERA5-Land overestimate SWE. ERA5-Land shows larger overestimation, which is mostly due to very high SWE values over mountainous regions. The analysis revealed a discontinuity in ERA5 around year 2004, since adding IMS (Interactive Multisensor Snow and Ice Mapping System) from year 2004 onwards considerably improves SWE estimates but makes the trends less reliable. The negative NH SWE trends in ERA5 range from -249 Gt decade[-1] to -236 Gt

decade[-1] in spring, which is two to three times larger than the trends detected by the other datasets (ranging from -124 Gt decade[-1] to -77 Gt decade[-1]). Albedo estimates are more consistent between the datasets with a slight overestimation in ERA5 and ERA5-Land. SCE is accurately described in ERA5-Land, whereas ERA5 shows notably larger SCE than the satellite-based datasets. The negative trends in albedo and SCE are strongest in May, when albedo trend varies from -0.011 decade[-1] to -0.006 decade[-1] depending on the dataset. The negative SCE trend detected by ERA5 in May (-1.22 million km$^2$ decade[-1])

is about twice as large as the trends detected by other datasets (ranging from -0.66 million km$^2$ decade[-1] to -0.50 million km$^2$ decade[-1]). The analysis also shows that there is a large spatial variability in the trends, which is consistent with other studies.

## 1 Introduction

Seasonal snow cover of the Northern Hemisphere (NH) is an important part of the global climate system. In winter, a large fraction of incoming solar radiation is reflected back due to the high albedo of snow cover, whereas in summer, the darker



snow-free surface absorbs more incoming solar radiation (Callaghan et al., 2011; Flanner et al., 2011; Qu and Hall, 2005; Trenberth and Fasullo, 2009). Therefore, changes in snow cover will affect the surface albedo and, thus, the surface energy balance.

Albedo is the hemispherical reflectance of the surface, defined as the ratio between reflected and incoming solar radiative
fluxes (Ångström, 1925). The solar radiation that reaches the Earth's surface can be either direct or diffuse (Schaepman-Strub et al., 2006). When incoming solar radiation propagates through atmosphere, it can be scattered to different directions due to, for example, clouds and aerosols. A part of this scattered radiation reaches the surface, and it is called the diffuse solar radiation. The solar radiation that reaches the surface without being scattered, is called direct solar radiation. Albedo can be divided into different components based on the solar radiation: The surface albedo, including both direct and diffuse
solar radiation, is often referred as blue-sky albedo. The albedo for direct solar radiation is the directional-hemispherical reflectance, often also called black-sky albedo. The albedo for the diffuse radiation, in turn, is bi-hemispherical reflectance and called white-sky albedo (Schaepman-Strub et al., 2006). The blue-sky albedo is a weighted average of black-sky and white-sky albedos (Schaepman-Strub et al., 2006). Clouds affect the ratio between the diffuse and direct solar radiation and, therefore, the changes in cloudiness also affect the blue-sky albedo (Key et al., 2001). Typically, the values for white-sky
albedo are larger than the values for black-sky albedo, and blue-sky albedo is somewhere between these two (Manninen et al., 2012; Manninen et al., 2019; Schaepman-Strub et al., 2006).

Snow cover affects many human activities, such as road traffic, tourism, forestry, and agriculture (Callaghan et al., 2011). In the high latitudes and in the mountainous regions, snow cover also greatly influences the hydrological cycle (Barnett et al.,
2005; Bormann et al., 2018; Callaghan et al., 2011; Douville et al., 2002; Li et al., 2017). In winter, snow cover stores a large amount of fresh water and limits water availability. When snow melts in spring and summer, water is released and redistributed. Melting snow is an important source of fresh water, as about one-sixth of the world's population depends on melt water from snow (Barnett et al., 2005). Hydropower production is also dependent on meltwater from snow (Callaghan et al., 2011; Magnusson et al., 2020). In many regions, most of the annual inflow to hydropower reservoirs often comes
during the spring snowmelt period (Magnusson et al., 2020). Therefore, changes in snow cover can cause both water shortages and affect hydropower production.

The global air temperature is rising due to climate change which causes the melt season to begin earlier and affects the timing of the streamflow peaks (Kundzewicz et al., 2008; Musselman et al., 2021). Rising temperatures also cause the winter
precipitation to shift from snow to rain, which affects the intensity of the streamflow during melt season (Kundzewicz et al., 2008). The streamflow is projected to decrease especially in USA and southern and central Europe (van Vliet et al., 2016). For example, in USA, snow is the largest source of water storage in the arid western states (Hall et al., 2008; Li et al., 2017), where the Colorado River, which originates from the Rocky Mountains, highly depends on the melt water from snow (Li et

al., 2017). The river provides drinking water to 40 million people and irrigates over five million acres of agricultural land
across seven states in the USA and two in Mexico (Smith et al., 2022). The warming climate will cause snowpack losses in
the Rocky Mountains, which will, in turn, affect the availability of water in Colorado River basin states (Hall et al., 2008;
Kundzewicz et al., 2008, Li et al., 2017).

Studies have shown that seasonal snow cover is changing, and snow cover in spring is especially sensitive to warming
(Derksen and Brown, 2012). In winter, there is not much sunlight in the Arctic, so changes in snow cover have a smaller
effect on the surface energy budget. In spring, the amount of incoming solar radiation increases, which also makes the
surface albedo feedback (SAF) stronger. Since there is still a lot of snow in the Arctic in the spring season, changes in the
spring snow cover greatly affect the surface energy balance and, thus, the climate system (Déry and Brown, 2007).

Snow water equivalent (SWE) is the amount of water that would result, if the snowpack would melt instantaneously, and it
can be expressed in the units of mm or, equivalently, in the units of kg m$^{-2}$ (Fierz et al., 2009). Recent studies show negative
trends in global SWE and in the extent and duration of NH snow cover, but seasonal and spatial variability exists (Bormann
et al., 2018; Derksen and Mudryk, 2022; Hernández-Henríquez et al., 2015; Pulliainen et al., 2020). In early winter from
October to December, most datasets show negative trends in snow cover extent (SCE), while in January and February, there
are no significant trends (Mudryk et al., 2017). The observed snow cover trends in spring are clearly negative in both Eurasia
and North America (Derksen and Brown, 2012; Essery et al., 2020; Hernández-Henríquez et al., 2015). Also, a clear trend
exists towards earlier melt season (Metsämäki et al., 2018; Takala et al., 2009; Tedesco et al., 2009). SWE shows large
spatial variability; in North America, there is a negative trend in observed SWE, whereas in Eurasia the trends are less
prominent. In Siberia, there are also regions where SWE is observed and projected to increase (Pulliainen et al., 2020;
Räisänen, 2008).

Particularly in the Arctic regions, snow cover plays a significant role in the climate system, making it crucial to understand
the characteristics of the snow cover in these areas. As in situ measurement network is relatively sparse, snow cover
monitoring at continental scale is only possible from satellites or using reanalysis data. Both methods can cover large areas
and provide snow cover estimates also in those regions which lack in situ observations and are therefore widely used in
climate research (e.g., Mortimer et al., 2020; Mudryk et al., 2020).

Reanalyses provide decadal time series with multiple variables making them suitable for various kind of research, and they
have become an important part of climate studies. ERA5 is the fifth generation ECMWF (European Centre for Medium-
Range Weather Forecasts) atmospheric reanalysis replacing the older ERA-Interim reanalysis (Hersbach et al., 2020). ERA5
uses advanced modeling and data assimilation systems and combines large amounts of historical observations into global
estimates. ERA5-Land, in turn, is the land component from ERA5 with a finer spatial resolution (Muñoz-Sabater et al.,



2021). It is produced without coupling the atmospheric module, and it runs without data assimilation making it computationally lighter (Muñoz-Sabater et al., 2021).


Snow cover properties in ERA5 and ERA5-Land have been evaluated in several studies (e.g., Mortimer et al., 2020; Urraca and Gobron, 2021; Räisänen, 2023). Overall, ERA5 and ERA5-Land have been found to perform well compared to other reanalyses (Jia et al., 2022; Lei et al., 2022; Mortimer et al., 2020). However, studies have shown that ERA5 tends to overestimate snow cover extent, SWE, and surface albedo (Bian et al., 2019; Guo and Yang, 2022; Orsolini et al., 2019;

Xiaona et al., 2020). Also, ERA5-Land shows a slight overestimation in surface albedo during snow-free season (Jia et al., 2022). ERA5 slightly underestimates very deep snow especially late in the snow season and shows delayed ablation of deep snowpack in spring (Lei et al., 2022; Mortimer et al., 2020). ERA5-Land SWE estimates agree well with other reanalysis datasets but struggles in comparison with satellite-based data (Räisänen, 2023). ERA5-Land shows especially large overestimation over mountainous areas (Monteiro and Morin, 2023). Also, a discontinuity in 2004 has been observed in

ERA5 snow cover estimates, which is due to adding IMS information in ERA5 (Urraca and Gobron, 2021; Mortimer et al., 2020). This discontinuity has improved the accuracy of snow cover estimates after year 2004 considerably, but decreased the temporal stability (Urraca and Gobron, 2021).

Even though several studies exist on evaluating either ERA5 or ERA5-Land, few studies exist on comparing snow cover

properties in ERA5 and ERA5-Land. Lei et al. (2022) studied the effect of spatial resolution on snow depth estimates over the Tibetan Plateau and found that the finer-resolution ERA5-Land is more consistent with in situ measurements than ERA5. Li et al. (2022) found that both ERA5 and ERA5-Land tend to overestimate snow depth at high elevation in Central Asia. Monteiro and Morin (2023) have assessed the performance of several reanalyses, including ERA5 and ERA5-Land, over the European Alps and concluded that ERA5-Land considerably overestimates SWE, while ERA5 is better in line with the

reference data. Urraca and Gobron, (2021) studied snow cover duration and the stability of the trends in ERA5 and ERA5-Land over NH but did not consider other variables related to snow cover. Thus, our study is to the authors' knowledge the first, where NH snow cover properties (SWE, SCE, and albedo) are compared between ERA5 and ERA5-Land and evaluated with several satellite-based datasets.

## 2 Data and Methods

### 2.1 ERA5 and ERA5-Land

The datasets used in this study are listed in Table 1. ERA5 is the fifth generation ECMWF atmospheric reanalysis replacing the older ERA-Interim (Hersbach et al., 2020). ERA5 covers years from 1950 onwards with a grid resolution of 31 km, which is a considerably higher spatial resolution than in ERA-Interim (80 km). ERA5 is based on Integrated Forecasting System (IFS) Cycle 41r2 and it provides hourly fields for all variables. Additionally, it provides pre-computed monthly





means (Hersbach et al., 2020), which we used in this study. The number of observations assimilated in ERA5 increases
constantly throughout the production period. ERA5 assimilates snow depth information from several SYNOP stations, and
from year 2004 onwards, it also uses IMS data over NH (Hersbach et al., 2020).

ERA5-Land, in turn, is a replay of the land component of ERA5 with a finer spatial resolution (9 km) (Muñoz-Sabater et al.,
2021). It is produced with land model H_TESSEL and without coupling the atmospheric module. Also, ERA5-Land runs
without data assimilation, which makes it computationally lighter (Muñoz-Sabater et al., 2021).

Our study includes three variables: SWE, SCE, and albedo. SWE was directly available in both ERA5 and ERA5-Land
(variable "sd"). In ERA5-Land, snow cover fraction in a grid cell (SC) is directly available (variable "snowc"), but for
ERA5, we calculated SC in a grid cell using snow density and SWE:

$$SC = \frac{\rho_w \cdot SWE\ [m]}{\rho_{snow}} \cdot \frac{1}{0.1} \qquad\qquad (1)$$

where $\rho_w$ is the density of water (1000 kg m$^{-3}$) and $\rho_{snow}$ is the density of snow. We calculated SCE by multiplying snow
cover fraction in a grid cell (SC) with the size of the grid cell.

ERA5 and ERA5-Land provide several estimates for albedo. Both ERA5 and ERA5-Land have directly available the albedo
for diffuse radiation, i.e., the white-sky albedo (variable "fal"). Also, ERA5 and ERA5-Land provide radiation estimates
from which blue-sky albedo can be computed. Additionally, ERA5 has radiation estimates for clear-sky conditions from
which blue-sky albedo for clear-sky can be calculated. Thus, we calculated the blue-sky albedo ($\alpha_{blue}$) using downward solar
radiation (SW$_{dn}$; variable "ssrd") and net solar radiation (SW$_{net}$; variable "ssr"):

$$\alpha_{blue} = 1 - \frac{SW_{net}}{SW_{dn}} \qquad\qquad (2)$$

Similarly, blue-sky albedo for clear-sky ($\alpha_{clear}$) was calculated using Eq. (2) with the radiation estimates for clear skies
(downward solar radiation for clear skies, variable "ssrdc", and net solar radiation for clear skies, variable "ssrc").

We compared the timeseries of all these albedo estimates (Fig. S1) and the analysis showed that the different albedo
estimates are very close to each other in both ERA5 and ERA5-Land. The satellite-based products (Sect 2.2) provide
estimates for black-sky albedo, which is not available in ERA5 or ERA5-Land. From the variables shown in Fig. S1, the
blue-sky albedo for clear-sky ($\alpha_{clear}$) would typically be closest estimate to black-sky albedo, whereas white-sky albedo is
typically the furthest away from black-sky albedo. However, the blue-sky albedo for clear-sky ($\alpha_{clear}$) is only available in





ERA5 and not in ERA5-Land. Therefore, we decided to use the blue-sky albedo ($\alpha_{blue}$) in this analysis, as it is available in both ERA5 and ERA5-Land. This issue is further discussed in Sect 2.2.


**Table 1. Datasets used in this study.**

| Variable | Dataset | Resolution | Reference |
|---|---|---|---|
| SWE<br>$SW_{dn}$<br>$SW_{net}$<br>$SW_{dn,clear}$<br>$SW_{net,clear}$<br>$\rho_{snow}$ | ERA5 | 31 km × 31 km, monthly | Hersbach et al. (2020) |
| SWE<br>SCE<br>$SW_{dn}$<br>$SW_{net}$ | ERA5-Land | 9 km × 9 km, monthly | Muñoz-Sabater et al. (2021) |
| SWE | Bias-corrected SnowCCI v1 | 25 km × 25 km, monthly | Luojus et al. (2021) |
| SWE | Brown | 0.75° × 0.75°, monthly | Brown et al. (2003) |
| SWE | Crocus v7 | 0.5° × 0.5°, monthly | Brun et al. (2013) |
| SWE | MERRA-2 | 0.5° × 0.625°, monthly | Gelaro et al. (2017)<br>GMAO (2015) |
| SCE | Rutgers | 24 km × 24 km, weekly | Robinson and Estilow (2021) |
| SCE | JAXA JASMES | 5 km × 5 km, half-monthly | Hori et al. (2017) |
| Black-sky albedo | CLARA-A2 SAL | 0.25° × 0.25°, monthly | Anttila et al. (2016)<br>Karlsson et al. (2017) |
| Black-sky albedo | MCD43D51 | 30 arcsec × 30 arcsec, daily | Schaaf and Wang (2021a) |
| Quality information | MCD43D31 | 30 arcsec × 30 arcsec, daily | Schaaf and Wang (2021b) |

## 2.2 Satellite-based datasets

The SWE reference data used in this study consists of four datasets: ESA CCI Snow "SnowCCI" (European Space Agency

Climate Change Initiative, Snow) v1 data (Luojus et al., 2021), MERRA-2 (the Modern-Era Retrospective analysis for Research and Applications, Version 2; Gelaro et al., 2017; GMAO, 2015a), Brown (Brown et al., 2003), and Crocus v7 (Brun et al., 2013).



SnowCCI v1 is the same product as the GlobSnow v3 SWE product except provided in geographical latitude-longitude grid
for easier comparison with climate model data. The product is based on satellite-based microwave brightness temperature
data and ground-based snow depth measurements (Luojus et al., 2021). Subsequently, the dataset is bias-corrected with
extensive ground-based snow course measurements, which decrease the uncertainty of hemisphere-mean SWE estimates
notably (Pulliainen et al., 2020). We have used SnowCCI v1, even though there is already version 2 available. Version 2 (v2)
has been shown to underestimate SWE and currently, the bias-corrected SnowCCI v1 is considered to be the best SWE
estimate when analyzing decadal timeseries (Mortimer et al., 2022). A notable difference between v1 and v2 is that v1 uses
constant snow density, while v2 uses spatially and temporally varying snow densities (Mortimer et al., 2022). The dynamic
snow density decreases the annual maximum SWE, which leads to underestimation of SWE in v2. Therefore, v1 currently
provides more reliable SWE estimates, and the bias-corrections further improve v1 making the bias-corrected SnowCCI v1
currently the best SWE estimate (Mortimer et al., 2022). The SnowCCI data are available for the period 1979-2018 and are
mapped to a 25 km EASE-Grid (Luojus et al., 2021).

As the SnowCCI data are only available for non-mountainous regions, we have used the mean of the MERRA-2, Brown, and
Crocus v7 SWE products for the mountainous regions. MERRA-2 is a NASA (National Aeronautics and Space
Administration) atmospheric reanalysis, and it is available from year 1980 (Gelaro et al., 2017). Brown SWE product, in
turn, uses a simple snow scheme driven by ERA-Interim reanalysis (Brown et al., 2003). Crocus version 7 product is a
physical snow model driven by ERA-Interim reanalysis (Brun et al., 2013). Both MERRA-2 and Crocus v7 tend to slightly
overestimate SWE under 150 mm and underestimate SWE over 150 mm (Mortimer et al., 2020).

We used two satellite-based surface albedo products in this study. CLARA-A2 SAL (Clouds, Albedo and Radiation second
release Surface Albedo) product provides the broadband shortwave directional-hemispherical reflectance, i.e., the black-sky
albedo (Anttila et al., 2016a; Karlsson et al., 2017). The product is based on AVHRR (Advanced Very High Resolution
Radiometer) data, and the data are available for the period 1982-2019. We used the monthly mean values with spatial
resolution of $0.25° \times 0.25°$ on a regular latitude-longitude grid. The mean relative retrieval error is -0.6%, the mean root
mean square error is 0.075 and the decadal relative stability (over Greenland Summit) is 8.5% (Anttila et al., 2016b). The
product has been found to perform especially well over snow and ice, which makes it well-suited for cryospheric studies
over the Arctic (Anttila et al., 2016b).

Additionally, we used the MCD43D51 product which is the black-sky albedo for the MODIS shortwave broadband (Schaaf
and Wang, 2021a). It is a daily product with a grid resolution of 30 arc second (1 km) and available from year 2000 onwards.
We also used the quality information (MCD43D31) for the albedo product (Schaaf and Wang, 2021b) and only included
pixels with good quality in the analysis, which ensures the high accuracy of the product.

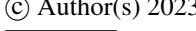



The satellite-based albedo products provide black-sky albedo estimates, whereas ERA5 and ERA5-Land albedo estimates are for blue sky-albedo, which can cause a slight uncertainty in the comparison. ERA5 would also provide blue-sky albedo

for clear-sky, which would typically be closer to black-sky albedo. However, that is not available in ERA5-Land, so we decided to use the blue-sky albedo to maintain consistency between ERA5 and ERA5-Land. Also, the comparison of the different albedo estimates in ERA5 and ERA5-Land showed that the differences between the different albedo estimates is negligible (Fig. S1). Studies have also shown that the difference between black-sky and blue-sky is less than 5 % with low aerosol optical depth (AOD) values (Manninen et al., 2012), which is typical for the Arctic region (Shikwambana and

Sivakumar, 2018). Therefore, we decided not to make any adjustments to any of the albedo datasets in order to minimize the possible discrepancies between black-sky and blue-sky albedo estimates. This approach has been used also in previous studies (e.g., Riihelä et al., 2013; Pohl et al., 2020). For simplicity, from now on, we will simply refer to albedo and do not differentiate between different types of albedos unless the difference is relevant.

The SCE products used in this study are the Rutgers weekly product (Robinson and Estilow, 2021) and JAXA JASMES SCE product (Hori et al., 2017). The Rutgers product provides weekly SCE at 24 km resolution. It is a binary product (no snow or snow-covered), and the threshold for which a grid cell is considered snow covered is 50% or greater (Robinson and Estilow, 2021). The data are available from year 1980 onwards. The Rutgers weekly SCE product at 190 km resolution has been widely used in climate research (e.g., Brown et al., 2010; Déry and Brown, 2007; Hernández-Henríquez et al., 2015), as the

dataset is available from year 1966 onwards making it the longest running satellite-based SCE record (Estilow et al., 2015). Recently, the product was gridded to a finer resolution, as the weekly SCE maps spanning from 1980 through 1999 were digitized at 24 km resolution. Since 1999, the SCE charts have been produced daily at 24 km resolution using the IMS data (Robinson and Estilow, 2021). Especially before IMS period, the data are most accurate with cloud-free conditions, stable or slow-changing snow cover and high solar illumination levels (Robinson and Estilow, 2021).


The JAXA JASMES product is based on AVHRR and MODIS data (Hori et al., 2017). The product provides the snow cover fraction (0-100%) at 5 km resolution and is available from year 1978 onwards. The product uses AVHRR before 2000 and MODIS data from year 2000 onwards (Hori et al., 2017). Studies have shown that JAXA JASMES tends to slightly overestimate SCE and the overestimation increases in spring. However, the overall accuracy of the dataset is very high (Hori

et al., 2017). Rutgers shows higher SCE than JAXA JASMES in the Arctic (north of 60°N) in May (Derksen and Mudryk, 2022).

**2.3 Methods**

We have used the nearest neighbor method to resample ERA5, ERA5-Land, Brown, Crocus, MERRA-2, Rutgers and JAXA JASMES to 25 km equal-area projection. The higher-resolution MODIS albedo product was first coarsened to 0.25°

resolution by calculating the mean value of all the grid cells within one 0.25° × 0.25° grid cell and subsequently resampled to 25 km equal-area projection using the nearest neighbor method. We have used monthly mean values for each variable. The snow cover extent (SCE) was calculated by multiplying the area of one grid cell (25 km × 25 km) with the snow cover fraction in each grid cell. Our study covers land areas north of 40° N (glaciers and ice sheets are excluded) and the period between 1982-2018 (spring season from March to May). As MODIS data are available only from year 2000 onwards, we

have done some analysis additionally for the period 2000-2018.

To study the trends, we used the Theil-Sen estimator (Sen, 1968; Theil, 1950), which is the median of all slopes between paired values, and it is less sensitive to outliers than the ordinary least squares linear regression. We calculated trends for the entire study area (Sect. 3.1) and separately for Eurasia and North America (Sect. 3.2). Additionally, we computed the trends

in each grid cell (Sect. 3.3) to study the spatial variability of the trends. If there were missing values when calculating the trends in each grid cell, we calculated trends only in grid cells with at least 15 values available during the study period (i.e., at least 15 years of data).

## 3 Results

Figure 1 shows as an example the mean values for each of the dataset in April 1982-2018 (MODIS 2000-2018). Overall, the

spatial distributions of SWE, albedo and SCE are quite similar in ERA5, ERA5-Land and the satellite-based datasets. SWE shows large spatial variability in all the datasets. The highest monthly mean values exceed 240 mm, and these are found in Rocky Mountains, southeastern Canada, Scandinavia and in Siberia. However, in ERA5 and ERA5-Land, the areas with very high SWE values are more extensive than in the SWE reference data, indicating that both ERA5 and ERA5-Land overestimate SWE compared to the SWE reference data. Also, ERA5-Land seems to overestimate SWE even more than

ERA5.

Albedo shows highest values in the northern part of the study area, and the spatial distribution is similar to the SWE distribution. All the albedo datasets are quite consistent with each other. When comparing ERA5 and ERA5-Land, ERA5-Land shows slightly higher values than ERA5. Spatial distribution of SCE is consistent between ERA5-Land, Rutgers and

JAXA JASMES, whereas ERA5 shows clearly larger SCE.

### 3.1 Timeseries and trends in NH

Both ERA5 and ERA5-Land notably overestimate SWE sum over the entire study area in every month in spring compared to the SWE reference data (Fig. 2, top row). ERA5-Land shows even higher values than ERA5, which was evident also in Fig. 1. The magnitude of the difference stays about the same (Fig. 2, second row) throughout the spring season indicating

that in late spring the relative difference is very large. The values in ERA5 and ERA5-Land are about two times higher in





March and almost three times higher in May compared to the SWE reference data. The difference between ERA5-Land and SWE reference data stays around 3000 Gt throughout the study period. For ERA5, in turn, there is a clear drop in the difference around year 2004, which is the year when IMS data was added to the reanalysis. The drop is more visible in March and April than in May.

Mean values in April 1982-2018



**Figure 1. Mean values in April 1982-2018 for each of the datasets (MODIS 2000-2018).**



**Figure 2. Timeseries in the whole study area in (left) March, (middle) April and (right) May for SWE sum (top row), difference in SWE sum (second row), albedo (third row) and SCE (bottom row).**

The timeseries for non-mountainous regions are shown in the Supplementary material (Fig. S2). The SWE values between the datasets are more consistent with each other, indicating that most of the overestimation in SWE occurs in the mountainous regions. Especially ERA5 is very well in line with the satellite-based dataset after year 2004.

The albedo timeseries (Fig. 2, third row) are more consistent with each other, which was already seen in Fig. 1. There is a slight overestimation in albedo in both ERA5 and ERA5-Land compared to CLARA-A2 SAL and MODIS data, but the





overestimation is not as prominent as in SWE. Even though ERA5 and ERA5-Land slightly overestimate albedo, both are

able to capture the annual variability quite well.

The SCE timeseries (Fig. 2 bottom row) show similar values for ERA5-Land, Rutgers and JAXA JASMES. ERA5, in turn,

show considerably larger SCE, which is consistent with Fig. 1. There is a slight drop in ERA5 SCE estimates in year 2004,

but it is not visible in albedo timeseries. Also, the timeseries of albedo and SCE for the non-mountainous regions (Fig. S2)

do not differ much from Fig 2., which is most likely due to the fact that even though mountain areas store a considerable

portion of snow mass (Kim et al., 2021), they only represent a small fraction of the whole study area. Therefore, they have

limited effect on SCE and albedo.

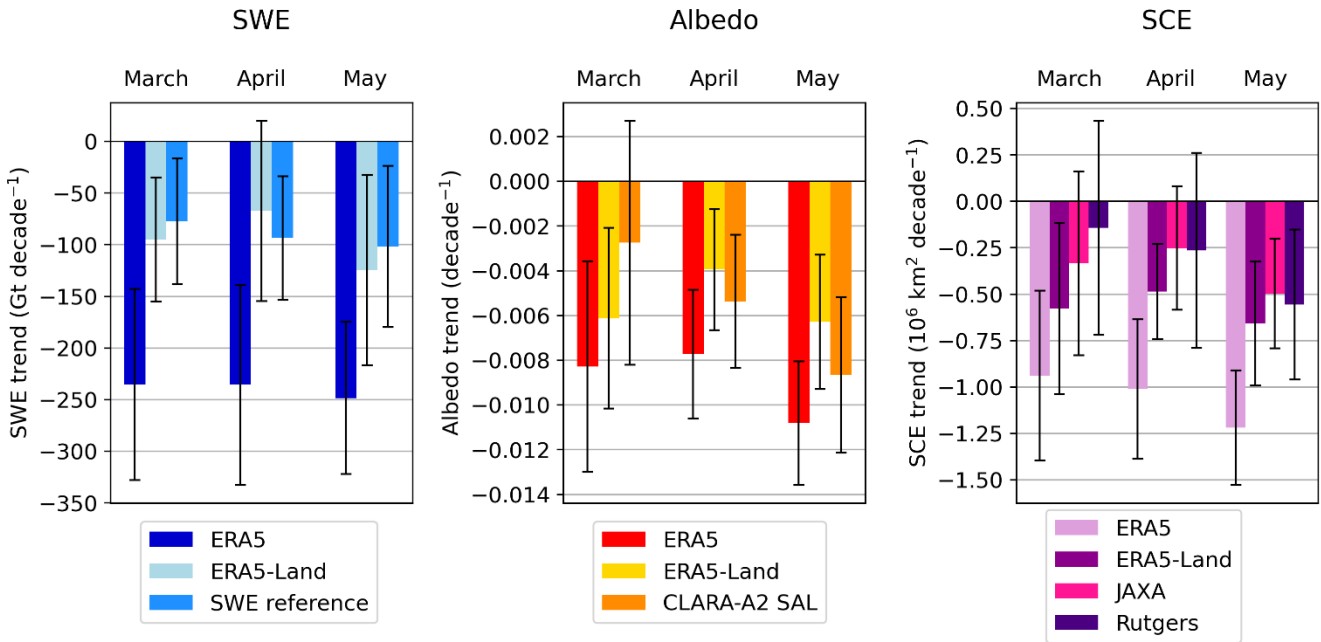

**Figure 3. Trends in Northern Hemisphere in 1982-2018. The error bars represent the 95% confidence intervals.**

Figure 3 summarizes the decadal trends and the 95% confidence intervals of the trends for each of the dataset. ERA5 shows

clearly negative SWE trends in every month (ranging from -249 Gt decade[-1] to -236 Gt decade[-1]). The trends in ERA5-Land

and the SWE reference data are also negative but not as prominent (ranging from -124 Gt decade[-1] to -77 Gt decade[-1]), which

means that ERA5 shows two or even three times larger decreasing trend than ERA5-Land and the SWE reference data.

Overall, the trends in ERA5-Land and SWE reference data are more consistent with each other, compared to ERA5. In

March, the difference between the trends in ERA5 and SWE reference cannot be explained with the uncertainties. All



datasets show statistically significant negative trends in March and May. In April, ERA5 and SWE reference data show
statistically significant negative trends.

The trends in albedo are more consistent between all the datasets. In March, CLARA-A2 SAL indicates that there is no statistically significant trend, contrary to ERA5 and ERA5-Land. In April and May, all three datasets show statistically significant trends, and the negative trends are strongest in May, when they vary from -0.011 decade$^{-1}$ to -0.006 decade$^{-1}$.
Consistent with SWE, ERA5 shows more prominent trends in each month than ERA5-Land and CLARA-A2 SAL.

The trends in SCE show large variability between the datasets. In March and April, both ERA5 and ERA5-Land show statistically significant negative trends, whereas the satellite-based JAXA JASMES and Rutgers indicate that there is not a statistically significant trend. In May, all the datasets show statistically significant negative trends, but the negative trend
detected by ERA5 (-1.22 million km$^2$ decade$^{-1}$) is about twice as large as the trends detected by other datasets (ranging from -0.66 million km$^2$ decade$^{-1}$ to -0.50 million km$^2$ decade$^{-1}$). Climate warming affects the snow cover in late spring and summer the most, while the changes in snow cover in late winter and early spring are minor (Derksen and Brown, 2012; Mudryk et al., 2017). This also affects the uncertainties, as the wide uncertainty range in March clearly decreases towards April and May. Similar to SWE and albedo, ERA5 shows the most prominent SCE trends in each month.

**3.2 Timeseries and trends in North America and Eurasia**

Figures 4 and 5 show the timeseries of SWE sum, albedo and SCE in North America and Eurasia. In North America, ERA5 and ERA5-Land show considerably larger values than the SWE reference data (Fig. 4, top row). The magnitude of the SWE difference stays about the same throughout the spring season, which is consistent with Fig. 2. A drop in the difference between ERA5 and the SWE reference data can be observed also in North America. In Eurasia (Fig. 5, top row), ERA5-Land
overestimates SWE considerably, whereas ERA5 is better in line with the SWE reference data. There is a clear difference between ERA5 and SWE reference data in the beginning of the study period, but the difference decreases throughout the study period, and after year 2004, the difference drops close to zero.

The albedo and SCE timeseries in North America and Eurasia (Figs. 4 and 5) show similar features than the timeseries in the
entire study area (Fig. 2). ERA5 and ERA5-Land show a slight overestimation in albedo and ERA5 overestimates SCE in both North America and Eurasia.

Figures 6 and 7 summarize the decadal trends and the 95% confidence intervals in North America and Eurasia. In North America (Fig. 6), ERA5 shows statistically significant negative trends in SWE in every month (ranging from -70 Gt decade$^{-1}$
to -57 Gt decade$^{-1}$), whereas ERA5-Land indicates that there is no trend at all. The SWE reference data, in turn, shows negative trends in March (-77 Gt decade$^{-1}$) and May (-41 Gt decade$^{-1}$). All the albedo datasets indicate that there is no trend



in March or April, whereas in May, all the datasets show statistically significant negative trend (ranging from -0.009 decade$^{-1}$ to -0.006 decade$^{-1}$). SCE also shows no trends, except ERA5 in March (-0.14 million km$^2$ decade$^{-1}$) and May (-0.26 million km$^2$ decade$^{-1}$).


**Figure 4. Timeseries in North America in (left) March, (middle) April and (right) May for SWE sum (top row), difference in SWE sum (second row), albedo (third row) and SCE (bottom row).**

In Eurasia (Fig. 7), both ERA5 and ERA5-Land indicate a statistically significant negative trend in every month, whereas the SWE reference data detect no trend at all, which is consistent with previous studies (Pulliainen et al., 2020). For ERA5, the SWE trend ranges from -191 Gt decade$^{-1}$ to -180 Gt decade$^{-1}$ and for ERA5-Land from -105 Gt decade$^{-1}$ to -102 Gt decade$^{-1}$. Especially in March, the trend is very close to zero according to the SWE reference data and becomes more prominent towards May, when the trend is -65 Gt decade$^{-1}$. There is a considerable difference in trends between ERA5, ERA5-Land





and the SWE reference data, which cannot be explained even with the uncertainties. The trends in albedo and SCE look similar to the ones detected in the whole study area (Fig. 3). In March, CLARA-A2 SAL indicates that there is not a statistically significant trend, but in April and May, all three datasets show statistically significant negative trends (ranging from -0.005 decade$^{-1}$ to 0.011 decade$^{-1}$). ERA5 and ERA5-Land show statistically significant SCE trends in every month, whereas JAXA JASMES and Rutgers only detect negative trends in May. The negative SCE trends detected by ERA5

(ranging from -0.78 million km$^2$ decade$^{-1}$ to -0.99 million km$^2$ decade$^{-1}$) are about twice as large as the trends detected by other datasets (ranging from -0.53 million km$^2$ decade$^{-1}$ to -0.31 million km$^2$ decade$^{-1}$). Trends in both albedo and SCE intensify as spring progresses, which was also evident in the whole study area (Fig. 3). Overall, the trends are more prominent in Eurasia than in North America.



**Figure 5. Timeseries in Eurasia in (left) March, (middle) April and (right) May for SWE sum (top row), difference in SWE sum (second row), albedo (third row) and SCE (bottom row).**





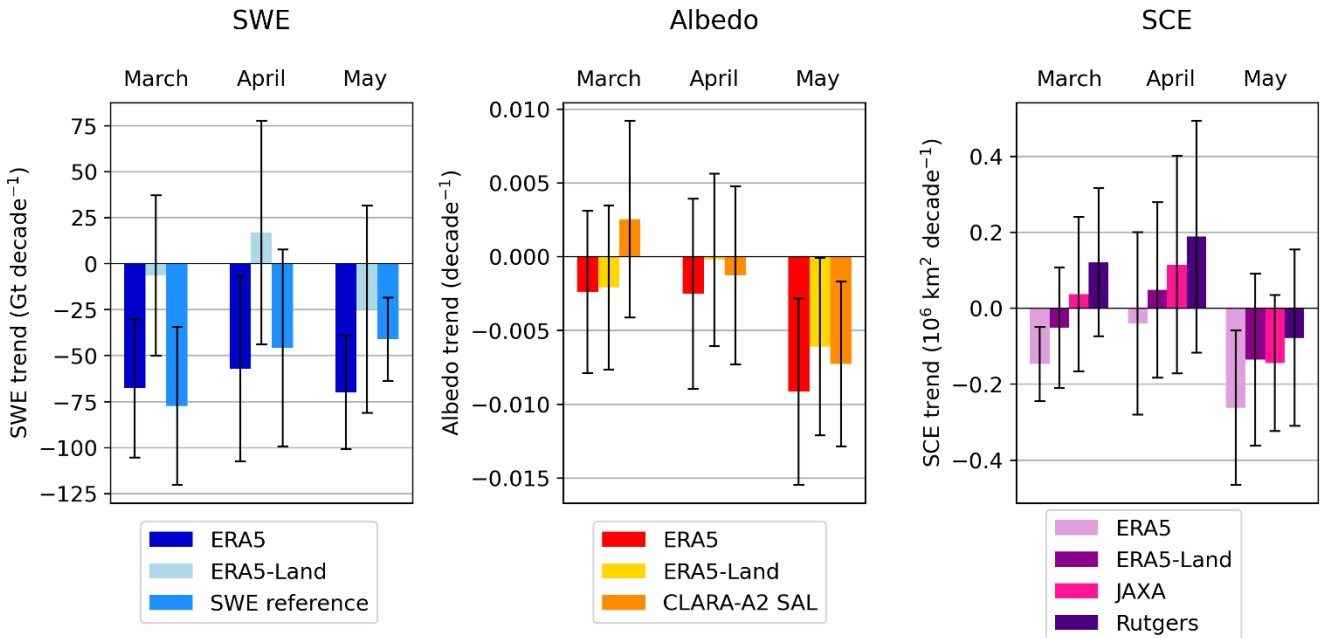

**Figure 6. Trends in North America in 1982-2018. The error bars represent the 95% confidence intervals.**


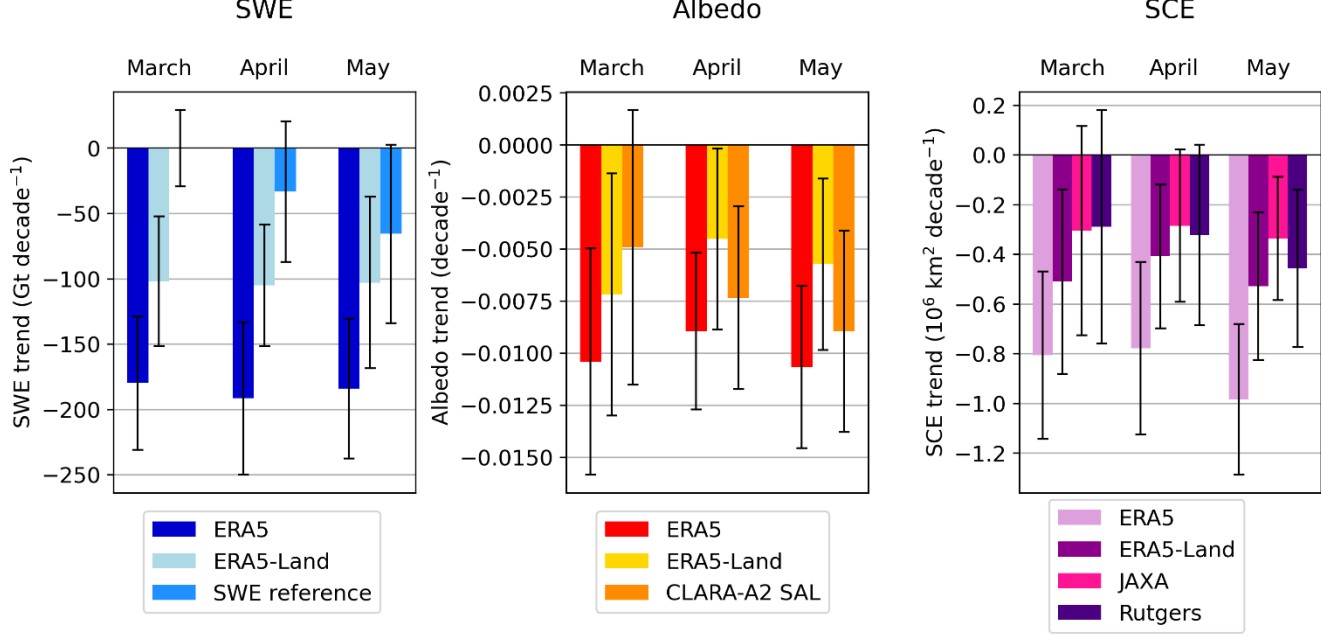

**Figure 7. Trends in Eurasia in 1982-2018. The error bars represent the 95% confidence intervals.**



### 3.3 Spatial trends in SWE and albedo

The trends in SWE for the period 1982-2018 show large spatial variability between the datasets (Fig. 8, top row). The trends in April and May are shown in the Supplementary material (Figs. S3-S4). ERA5 detects mostly negative trends in the whole study region in March. ERA5-Land, in turn, shows negative trends in the western side of Eurasia and very weak trends in North America and in eastern Eurasia. The SWE reference data detect negative trends in northern part of North America but shows large variability in trends in Eurasia. All datasets indicate a positive trend in Lapland. The trends in March in albedo

(Fig. 8, bottom row) are more consistent between the datasets. There is a large area in southwestern part of Eurasia with a clear negative trend related to loss of low-latitude seasonal snow cover in every dataset. A positive trend is visible in Canada; however, the size of the area shows some variability between the datasets. Also, CLARA-A2 SAL detects a positive trend in southeastern Eurasia, which is not visible in ERA5 or ERA5-Land.

We additionally studied the trends for the period 2000-2018, as MODIS data are only available from year 2000 onwards. The SWE trends for the period 2000-2018 (Fig. 9, top row) are clearly more consistent with each other, which is most likely due to adding the IMS information to ERA5, which improves SWE estimates after 2004. The trends in April and May are shown in the Supplementary material (Figs. S5-S6). Overall, the trends are more prominent in 2000-2018 than in 1982-2018. All datasets detect negative trends in northeastern Canada, west coast of North America and western Eurasia in March.

Positive trends, in turn, are shown over a large area in Canada, Scandinavia and in many areas in Siberia. The positive trend in Canada is consistent with other studies and it is associated with cooling in spring season (Mudryk et al., 2018). The positive trend is very strong in Scandinavia, and it is detected by all the datasets. We additionally analyzed in situ snow depth measurement across Lapland (Fig. S7) to investigate, weather the positive trend is also visible in in situ measurements. For the period 1982-2018, the trend based on in situ measurements is negligible, but for 2000-2018, there is a statistically

significant positive trend (9.3 cm decade$^{-1}$ in March), which is consistent with Figs. 8 and 9.

MODIS and CLARA-A2 SAL are consistent with each other, whereas ERA5 and ERA5-Land show some discrepancies in albedo trends. All datasets show a strong positive trend in albedo in Central Asia (fig. 9, bottom row), which is associated with an increase in snow cover over that area (Li et al., 2018). The albedo over Central Asia has been observed to increase

also in summer due to forestation (Li et al., 2018). Albedo has been increasing also over Canada, which is consistent with the positive SWE trend. Also, a negative trend is detected over Europe and western parts of Russia, where SWE has also been decreasing. Both ERA5 and ERA5-Land show a strong positive trend in northern Siberia, which is not as prominent in MODIS or CLARA-A2 SAL products. Also, MODIS and CLARA-A2 SAL show a negative trend in Central Siberia, which is not visible in ERA5 or ERA5-Land.





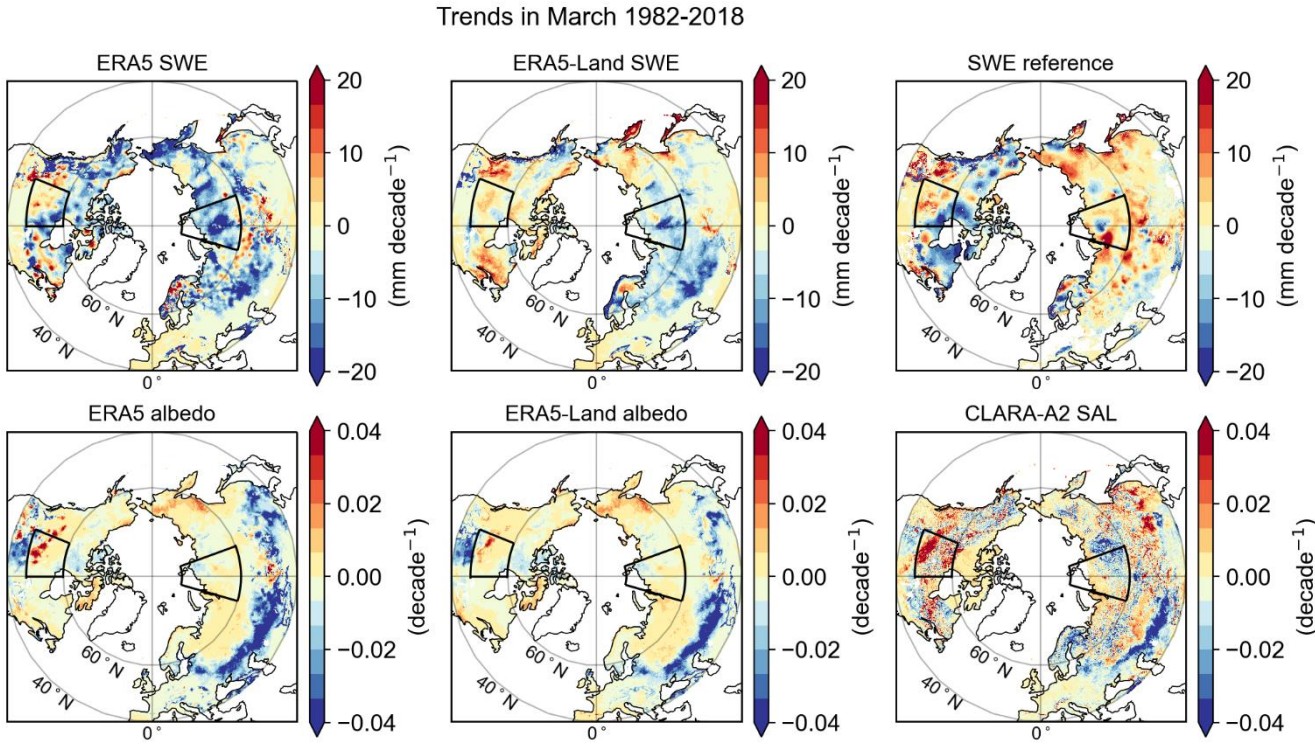


**Figure 8.** (top row) SWE trends in March 1982-2018. (bottom row) Albedo trends in March 1982-2018.

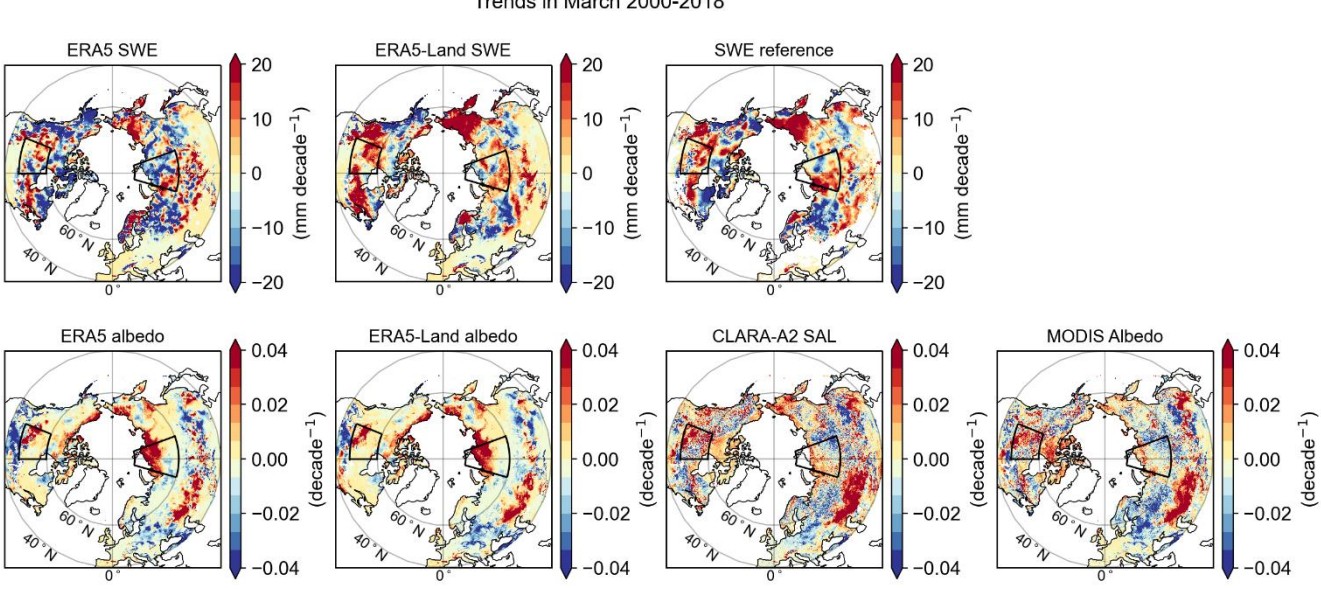

**Figure 9.** (top row) SWE trends in March 2000-2018. (bottom row) Albedo trends in March 2000-2018.



## 3.3 Regional trends

We additionally studied timeseries in smaller areas in the Western Siberia and the Canadian Prairies (areas marked in Figs 8 and 9). We chose these areas, because the trends in 1982-2018 show large differences between datasets but are more consistent in 2000-2018. In Siberia, both ERA5 and ERA5-Land overestimate SWE in the beginning of the study period (Fig 10, first and second row). The difference in SWE sum decreases throughout the study period and there is a clear drop in the difference between ERA5 and the SWE reference data in year 2004. After 2004, the difference is close to zero, with some annual variability. ERA5-Land does not show a clear improvement in 2004, but the difference is more stable throughout the study period.

**Figure 10.** Timeseries in Western Siberia in (left) March, (middle) April and (right) May for SWE sum (top row), difference in SWE sum (second row), albedo (third row) and SCE (bottom row).The area is marked in Figs. 8 and 9.





**Figure 11. Timeseries in Canadian Prairies in (left) March, (middle) April and (right) May for SWE sum (top row), difference in SWE sum (second row), albedo (third row) and SCE (bottom row). The area is marked in Figs. 8 and 9.**

Both ERA5 and ERA5-Land show a small overestimation in albedo (Fig 10, third row). The overestimation compared to satellite-based data is at its lowest in April. Even though there is a difference in the albedo values between ERA5, ERA5-Land and the satellite-based datasets, both ERA5 and ERA5-Land capture the annual variability quite accurately. All SCE datasets (Fig 10, bottom row) show that the whole area is covered with snow in March and April, with a few minor exceptions. In May, when the spring advances and melt season starts, the variability increases. ERA5 shows overestimation is SCE compared to other datasets, while ERA5-Land is very well in line with JAXA JASMES and Rutgers.

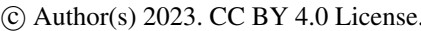

In the Canadian Prairies, the SWE estimates are quite consistent with each other (Fig. 11, first and second row). There is only a minor variability between the datasets and the difference varies from negative to positive. Especially in May, both ERA5 and ERA5-Land show mostly smaller SWE estimates than the SWE reference data. Contrary to Western Siberia, there is no clear drop in SWE sum difference in year 2004.

All the albedo datasets are very consistent in the Canadian Prairies in March, but when spring advances, the difference increases (Fig. 11, third row); in May, both ERA5 and ERA5-Land show a slight overestimation in albedo. Also, in SCE (Fig. 11, bottom row), the variability between the datasets increases when spring advances. ERA5 overestimates SCE also in this area throughout the spring season, and the overestimation is greatest in May. ERA5-Land, in turn, shows smaller SCE values in March compared to Rutgers and JAXA JASMES. In April and May, ERA5-Land, Rutgers and JAXA JASMES are
mostly consistent with each other. Both Figs. 10 and 11 illustrate that even though there are discrepancies between ERA5, ERA5-Land and the satellite-based datasets, both ERA5 and ERA5-Land are able to capture the interannual variability quite accurately.

## 4 Discussion

ERA5 and ERA5-Land show overall too large SWE sum estimates, which are mostly concentrated in the mountainous
regions (Figs. 1 and 2). The SWE estimates in the non-mountainous regions are better in line with the SWE reference data (Fig. S2). This result is consistent with other studies, as snow depth above 1500 m has been found to be unrealistically large in ERA5 (Hersbach et al., 2020). ERA5 has also shown a delayed ablation of deep snowpack in spring in Tibetan Plateau (Lei et al., 2022). The snowpack is presented in IFS with a single layer of snow which does not produce enough melting, and this results in excessively high snow depths (Hersbach et al., 2020).


Also, studies have shown that ERA5-Land overestimates snow depths more than ERA5 over the mountains (Monteiro and Morin, 2023; Muñoz-Sabater et al., 2021), which is consistent with our study. Both ERA5 and ERA5-Land show too high values in mountainous regions, but ERA5 has been found to perform better at the highest mountains (>3300 m), whereas ERA5-Land show improvements on mid-altitude mountains due to the higher resolution (Muñoz-Sabater et al., 2021). This
also mostly likely explains, why ERA5-Land overestimates SWE more than ERA5 (Fig. 2). The comparison between North America and Eurasia (Figs. 4 and 5) showed that ERA5 and ERA5-Land overestimate SWE more in North America, which likely results from the very high SWE values in Rocky Mountains. As the highest elevations of the Rocky Mountains exceed 3300 m, ERA5-Land shows even larger overestimation in SWE compared to ERA5.

The regional analysis (Sect. 3.3) showed that there is a large regional variability between the datasets. Even though ERA5 and ERA5-Land show overall larger SWE estimates, the difference is mostly opposite in Canadian Prairies. ERA5 shows



smaller SWE estimates throughout spring season, whereas for ERA5-Land, the difference decreases and drops under zero when spring advances (Fig. 11). However, also in these smaller regions, the difference between ERA5 and ERA5-Land stays the same, as ERA5-Land show larger SWE estimates than ERA5.


The SCE timeseries between ERA5-Land and the satellite-based datasets show only minor differences, while ERA5 notably overestimates SCE. That is, ERA5 tends to overestimate SCE, while ERA5-Land tends to overestimate SWE. The overestimation of SCE in ERA5 occurs in regions with shallow snowpack, which means that it does not convert to excessively large SWE values. ERA5-Land, in turn, overestimates SWE especially over mountainous regions with deep

snowpack. These areas are typically completely covered with snow and, therefore, overestimating SWE does not cause excessively high SCE values.

Timeseries of albedo are overall quite consistent between the datasets. Both ERA5 and ERA5-Land show a slight overestimation in albedo (Fig. 2) and when spring advances the difference slightly increases. Also, the analysis shows that

ERA5-Land shows moderately higher albedo values than ERA5. The discrepancy between satellite-based datasets and the reanalysis can be due to the different albedos used in the analysis. The satellite-based datasets provide estimates for black-sky albedo, whereas ERA5 and ERA5-Land provide estimates for blue-sky albedo. The black-sky albedo only accounts for the direct solar radiation, while the blue-sky albedo is for both direct and diffuse solar radiation (Schaepman-Strub et al., 2006). Typically, blue-sky albedo shows higher values than black-sky albedo (Manninen et al., 2012), which may cause

discrepancies in the comparison.

The comparison between the different albedo estimates in ERA5 and ERA5-Land (Fig. S1) showed only a slight difference between the albedo estimates and indicates that the differences between albedo estimates are smaller than typically. Typically, the albedo over snow and ice is 4-6% (absolute) higher under cloud cover than clear skies (Key et al., 2001). Fig.

10 shows that in March, the study area in Western Siberia is fully covered with snow. Therefore, we additionally compared the ERA5 and ERA5-Land albedo estimates over that area. The difference between white-sky albedo and blue-sky albedo for clear-sky only showed 1.1% (absolute) higher values (ranging from 0.4% to 1.7%), which is notably smaller difference than typically. Studies have shown that radiation quantities are slightly biased in ERA5 (Babar et al., 2019; Urraca et al., 2018), which would also affect albedo.


Even though there are discrepancies between ERA5 and ERA5-Land and the satellite-based datasets, both ERA5 and ERA5-Land are mostly able to capture the annual variability quite accurately. This finding is also consistent with other studies (Orsolini et al., 2019). Especially the regional analyses (Figs. 10 and 11) show large annual variability, which both ERA5 and ERA5-Land are mostly able to capture. For example, in Western Siberia (Fig. 10), there is a clear increase in all the



variables in May after year 2011, which all the datasets detect similarly. In Canadian Prairies (Fig. 11) albedo shows large
interannual variability in March throughout the study period, which both ERA5 and ERA5-Land capture quite accurately.

The trends in 1982-2018 show considerable variability among the datasets (Fig. 3). ERA5 shows overall most prominent
trends in all variables. Especially the trend in SWE differs considerably from ERA5-Land and the SWE reference data.
Recent studies have shown that there is a discontinuity in snow cover estimates in 2004, which is caused by adding the IMS
information to ERA5 (Mortimer et al., 2020; Urraca and Gobron, 2021). This improves the SWE estimates after year 2004,
but it decreases the stability of the timeseries, and, therefore, makes the trends less reliable. ERA5-Land does not exhibit the
same discontinuity, which improves the stability of trends, but in turn, decreases the accuracy of snow cover estimates at the
end of the study period. The satellite-based datasets can also exhibit discontinuities due to using different satellite
instruments, but typically these are taken into account so that the datasets are suitable for climate studies (Hori et al., 2017;
Luojus et al., 2021).

IMS information is assimilated only on altitudes under 1500 m, so it does not improve snow cover estimates in mountainous
regions. This most likely explains, why the discontinuity is more visible in Eurasia than in North America (Figs. 4 and 5). As
the Rocky Mountains account for a large fraction of SWE sum in North America, adding IMS has smaller effect on SWE
sum estimate at continental scale. This phenomenon is also seen in the trends in North America and Eurasia (Figs. 6 and 7).
The SWE trends in North America are similar in ERA5 and the SWE reference data (Fig. 6), while there is a large difference
in Eurasia (Fig. 7). Since there is not a clear discontinuity visible in North America, it improves the stability of the trends
and makes them more reliable. The downside is that the SWE values itself are not improved towards the end of the study
period. In Eurasia, in turn, the trends show a considerate difference between ERA5 and the SWE reference data, which
cannot be even explained with the uncertainties. This suggests that the trends in 1982-2018 are not reliable, but the SWE
estimates notably improve after year 2004 and are very similar with the SWE reference data. This conclusion is consistent
with previous studies (Mortimer et al., 2020; Urraca and Gobron, 2021).

There are also uncertainties relates to the satellite-based datasets, which can affect the comparison. The satellite-based SWE
is more reliable than before, but, still, some uncertainties do exist. The comparison between JAXA JASMES and in situ
observations showed a slight overestimation in JAXA JASMES (Hori et al., 2017). Also, the difference between JAXA
JASMES and Rutgers increases towards May, indicating that melting snow may pose a challenge to satellite-based snow
detection. The uncertainty related to MCD43D51 product is overall very low, as we have only included the pixels with good
quality in the analysis. Also, CLARA-A2 SAL performs overall accurately over snow and ice (Anttila et al., 2016b), which
increases the reliability of this analysis.



In mountainous regions, the high topographic variability can cause uncertainties in the satellite-based estimates, as the relatively coarse resolution of satellite-data is not ideal for mountainous regions. The complex terrain causes uncertainties in
SWE estimates but averaging over multiple products can improve the accuracy (Mortimer et al., 2020). Mountains can also complicate albedo retrieval due to shadowing, but this is taken into account by making the topography correction for the CLARA-A2 SAL product (Anttila et al., 2016). Also, as mountains only represent a small fraction of the whole study area, they have limited effect on the overall albedo and SCE values.

**5 Conclusions**

We have evaluated snow cover properties in ERA5 and ERA5-Land and compared the timeseries and trends with several satellite-based datasets. Our study included SWE, SCE and albedo, which are the most important variables related to snow cover. Our study covers land areas north of 40 °N and the spring season from March to May in 1982-2018. The main findings in our study are as follows:

- Both ERA5 and ERA5-Land overestimate SWE, with ERA5-Land SWE estimates being the largest. The difference between ERA5-Land and SWE reference data remains at about 3000 Gt throughout the study period, which means that the NH total seasonal snow mass estimated by ERA5-Land is about two times higher in March and almost three times higher in May compared to the SWE reference data. The excessively high SWE is mostly due to the very large values over mountainous regions.

- There is a discontinuity in ERA5 around year 2004, since adding IMS from year 2004 onwards considerably improves SWE estimates but affects the temporal stability. ERA5-Land does not exhibit the same discontinuity, which improves the temporal stability of the trends, but in turn, decreases the accuracy of snow cover estimates at the end of the study period.

- Due to the discontinuity, ERA5 shows two or even three times larger decreasing trend than ERA5-Land and the
SWE reference data: SWE trends in ERA5 range from -249 Gt decade$^{-1}$ to -236 Gt decade$^{-1}$ in spring, while the trends in ERA5-Land and the SWE reference data range from -124 Gt decade$^{-1}$ to -77 Gt decade$^{-1}$ in spring.

- ERA5 and ERA5-Land albedo estimates are quite consistent with the satellite-based datasets with only a slight overestimation. However, this discrepancy may be explained with the discrepancy between the used variables: the satellite-based datasets provide estimates for black-sky albedo, whereas ERA5 and ERA5-Land provide estimates
for blue-sky albedo. Typically, the blue-sky albedo is slightly higher than the black-sky albedo, which may explain the difference. The negative trend in albedo is strongest in May, when it varies from -0.011 decade$^{-1}$ to -0.006 decade$^{-1}$ depending on the dataset.

- SCE is very accurately described in ERA5-Land, whereas ERA5 shows considerably larger values compared to the satellite-based datasets. Similar to albedo, the negative trends become more prominent when spring advances, and



in May, all the datasets show statistically significant negative trends. However, the negative trend detected by ERA5 (-1.22 million km$^2$ decade$^{-1}$) is about twice as large as the trends detected by other datasets (ranging from -0.66 million km$^2$ decade$^{-1}$ to -0.50 million km$^2$ decade$^{-1}$).

- Despite the discrepancies in ERA5 and ERA5-Land with the satellite-based datasets, both are able to capture the interannual variability quite accurately.

## Data availability

The bias-corrected SnowCCI data are available online at https://doi.org/10.1594/PANGAEA.911944. The Brown and Crocus v7 datasets are available from the original authors (please see Table 1 for references). MERRA-2 SWE data are available online at https://doi.org/10.5067/RKPHT8KC1Y1T. CLARA-A2 SAL data are available online at https://doi.org/10.5676/EUM_SAF_CM/CLARA_AVHRR/V002. MCD43D51 data are available online at https://ladsweb.modaps.eosdis.nasa.gov/archive/allData/6/MCD43D51/. Rutgers SCE data are available online at https://nsidc.org/data/g10035/versions/1. JAXA JASMES SCE data are available online at https://www.eorc.jaxa.jp/JASMES/index.html. ERA5 and ERA5-Land are available online at https://cds.climate.copernicus.eu.

## Author Contributions

KK conducted the analysis and drafted the manuscript. All authors contributed to manuscript review and editing.

## Competing interests

The authors declare that they have no conflict of interest.

## Acknowledgements

The work of KK has been funded by a personal grant from Väisälä Fund and by Academy of Finland (decision number 585    341845).

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
