# Peer review of "Evaluation of snow cover properties in ERA5 and ERA5-Land with several satellite-based datasets in the Northern Hemisphere in spring 1982-2018"

_The Cryosphere, 2023_

## Author Comment (AC1)

The review comments are shown in black and the author responses in red.

We would like to thank the reviewer for reviewing our manuscript. We appreciate all the comments and will revise our manuscript according to them. Please find below our responses to the comments.

This is a worthwhile study, although it is far from the first to evaluate model simulations of SWE, SCE and albedo in comparison with satellite-based datasets. It may be that this paper is the first with this precise combination of snow properties, reanalyses and observations, but it is mostly restricted to documenting differences without adding understanding. How much of the exaggerated SCE trend in ERA5 is due to the discontinuity in IMS assimilation? How closely are SCE and albedo anomalies related, and to what extent are they masked by forest cover? Difference plots for the means in Figure 1 would be useful. What can be said about the relative contributions of assimilation and resolution to differences between ERA5 and ERA5-Land?

We will edit the manuscript to address these questions and to add understanding about the differences. We will add more discussion on IMS and how adding IMS affects the snow cover estimates. We will discuss more closely the relationship between SCE and albedo. Also, we will add more information and discussion about the differences between ERA5 and ERA5-Land. We will also add difference plots for the means in Figure 1.

The abstract states that "The analysis shows that both ERA5 and ERA5-Land overestimate SWE". The three datasets used as the reference for SWE in mountainous regions are themselves model products, two of them driven by the ERA-Interim predecessor of ERA5. All that is conclusively shown here is that ERA5 and ERA5-Land have larger SWE in mountainous regions than these other model products. Does the overestimate of SCE in ERA5, even after 2004, just show that IMS overestimates SCE? How much of the differences in SWE be attributed to differences in precipitation and temperature driving data between ERA5 and ERA-Interim?

It is true that in mountainous regions, this analysis shows the difference between ERA5 or ERA5-Land and other model products. Currently, there are no observation-based SWE products available for mountainous regions. However, as mountain areas store a considerable portion of snow mass, we decided to include them in this analysis as well, despite the lack of observation-based data products. We acknowledge that comparing ERA5 and ERA5-Land with other model products might be problematic and therefore, we have also plotted the timeseries for non-mountainous regions only (Fig. S2). In the mountainous regions, we have used the mean SWE of three model products, which is an approach that has been used in other studies too (Mudryk et al., 2020; Derksen and Mudryk, 2023).

Also, averaging over multiple products can improve the accuracy of SWE estimates (Mortimer et al., 2020), making the SWE estimates more reliable.

However, we agree that this issue needs further discussion and therefore, we will add more discussion on this topic. We will also add discussion about the issue that model products use ERA-Interim and how this affects the analysis. Also, we will check the wording and will emphasize throughout the text, that in mountainous regions, we are comparing ERA5 or ERA5-Land with other model products and not with observation-based products so that the issue will be clear for the readers. We will also add more information and discussion about IMS and its effect on snow cover estimates.

Derksen, C. and Mudryk, L.: Assessment of Arctic seasonal snow cover rates of change, The Cryosphere, 17, 1431–1443, https://doi.org/10.5194/tc-17-1431-2023, 2023.

Mortimer, C., Mudryk, L., Derksen, C., Luojus, K., Brown, R., Kelly, R., and Tedesco, M.: Evaluation of long-term Northern Hemisphere snow water equivalent products, Cryosphere, 14, 1579-1594, https://doi.org/10.5194/tc-14-1579-2020, 2020.

Mudryk, L., Santolaria-Otín, M., Krinner, G., Ménégoz, M., Derksen, C., Brutel-Vuilmet, C., Brady, M., and Essery, R.: Historical Northern Hemisphere snow cover trends and projected changes in the CMIP6 multi-model ensemble, Cryosphere, 14, 2495-2514, https://doi.org/10.5194/tc-14-2495-2020, 2020.

Because discussion of hemispheric timeseries trends is followed by the same for continents, Figures 2 and 3 and much of the discussion in 3.1 could be cut.

After consideration, we decided to keep Sect 3.1, as we think it will bring useful information about the changes in snow cover. Our logic in the Results section is to start with large-scale results and move towards a smaller scale. We think it is useful to show the results also for the entire NH, as the changes in snow cover will affect, for example, the Earth's energy budget.

**Minor points:**

The albedo paragraph starting at line 34 interrupts the discussion of snow cover; I think it would sit better at a later point in the introduction.
We will move this paragraph to Sect. 2.1.

88 "relatively sparse" – relative to what?
We will remove the word "relatively".

110 Important to note here that IMS does not provide information on SWE, and it is not assimilated in ERA5 at elevations above 1500 m. Describe how assimilation of SCE is used to update SWE.

We will edit the text according to the comment.

136 This sentence is a repeat from line 98.

We will remove this sentence.

141 Note that Equation (1) is from the HTESSEL documentation (and needs to be limited to a maximum of one).

We will revise the text according to the comment and add a reference.

168 Not all of the datasets in 2.2 are satellite based.

We will change the subtitle to "Reference datasets".

184 Mortimer et al. (2022) referenced here did not evaluate the bias-correct SnowCCI and states that v2 is an improvement relative to v1.

It is true that there are improvements in v2 relative to v1. For example, v2 shows better seasonal evolution of SWE. However, using dynamic density in v2 also decreases SWE estimates, which are well below the SWE estimates from reanalysis products (Fig. 4 in Mortimer et al., 2022). Therefore, v1 is better when analyzing multidecadal timeseries and trends. Furthermore, the bias corrections improve the SWE estimates by increasing the general SWE level and thus correcting the SWE estimates closer to the real level. Therefore, the bias-corrected SnowCCI v1 is the most accurate SWE product for this analysis. We will revise the text to make the reasoning for using v1 instead of v2 clearer for the reader.

208-214 Availability of albedo estimates from ERA5 and ERA5-Land, and differences between them, have already been discussed in 2.1.

We will remove these sentences to avoid repetition.

261-265, Figure 1 Differences in SCE dominates differences in albedo, so should be discussed and shown first.

We will edit the text according to the comment.

Figure 2 Axis labels for the second row should show that this is change in SWE, not SWE.

We will edit the figure according to the comment.

393 "whether the positive trend"

We will edit the text according to the comment.

400 "deforestation"
We will edit the text according to the comment.

498-499 Is this intended to say that ERA5 and ERA5-Land are well correlated with observations of annual variability? That is not very obvious in Figures 10 and 11, but could be quantified.
Yes, this is what we intended to say. We will quantify this and will also edit the text so that this will be clear for the readers.

519 "the SWE values themselves"
We will edit the text according to the comment.

520 "a considerable difference"
We will edit the text according to the comment.

525 "uncertainties related to"
We will edit the text according to the comment.

The writing is generally good. I noted a number of incorrect commas, but the Finns have a word for reviewers who pay excessive attention to commas.
We will check the grammar throughout the text.

---

## Author Comment (AC2)

The review comments are shown in black and the author responses in red.

We would like to thank the reviewer for reviewing our manuscript. We appreciate all the comments and will revise our manuscript according to them. Please find below our responses to the comments.

In this manuscript, the authors presented an evaluation of ERA5 and ERA5-Land SWE, albedo, and SCE products with different satellite-based datasets in the NH during the spring from 1982-2018. While the study is not that innovative, the manuscript is comprehensive, well-written and of interest to the snow community and final users. However, I think that some restructuring of the Introduction is needed, and some issues need to be better discussed throughout the manuscript.

**Abstract**

I would appreciate seeing some quantitative results about the agreement among datasets rather than inserting all the information about trends, that is for sure useful but a little bit difficult to follow.
We will edit the abstract and will add some quantitative results about the differences between the datasets.

Also, when you mention "other datasets" (L20), it is not completely clear if you refer to the satellite-based datasets used as "reference" or if you are also accounting for differences between ERA5 and ERA5-Land.
We will reword this sentence to make it clearer.

L18 IMS first entire name than acronym
We will edit the text according to the comment.

**Introduction**

To make it easier to follow, I would always keep the same order as in the abstract, i.e., SWE, albedo and SCE (or maybe a different order that better suits for discussion). I think the introduction needs restructuring. First, you introduce the snow importance, hence L48-56 should be moved at the beginning. Secondly, you introduce SWE, albedo and SCE. Then you should introduce the reanalysis data and their importance also linked to climate change. However, L58-67 might be shortened. Finally, the aim of the work.

We will edit the introduction according to the comment. We will also edit the text and figures so that the order of the variables will always be the same.

It should also be clearly stated why you focus on the NH. Is that because of the lack of studies? Is a global evaluation of the snow properties already available for such a period? I suggest highlighting the importance of snow in the NH in general, despite mentioning the Arctic region throughout the text (as L86) that still belongs to the NH but it is just a part of that.

This analysis has concentrated only on NH because there is not much seasonal snow in Southern Hemisphere (SH). In NH, at its largest in winter, snow covers on average more than 45 million $km^2$, which is almost half of the NH land surface area (Estilow et al. 2015). Snow cover in the Southern Hemisphere (SH) accounts for less than 1% of SH land surface at its largest (Foster et al. 2009). As a consequence, the changes in SH snow cover are minor compared to the changes in NH and therefore, this study has concentrated only on the NH. We will edit the text to point out this issue.

Estilow, T. W., Young, A. H., and Robinson, D. A.: A long-term Northern Hemisphere snow cover extent data record for climate studies and monitoring, Earth Syst. Sci. Data, 7(1), 137-142., https://doi.org/10.5194/essd-7-137-2015, 2015.

Foster, J., D. Hall, R. Kelly and L. Chiu. Seasonal snow extent and snow mass in South America using SMMR and SSM/I passive microwave data (1979–2006)'. Remote Sensing of Environment 113:2, pp. 291–305, 2009.

**Minor comments**

L51 "limits water availability" is expressed in a negative way. I would use something like "stores water"
We will reword this sentence.

Provide short information about IMS and what kind of data is assimilated.
We will add information about IMS.

**Section 2**

L138 keep same order SWE, albedo, SCE
This will be applied throughout the text.

Eq. 1 is not completely clear to me. From the first term you get the snow height, that is then multiplied by 1/0.1 to derive the snow cover fraction. Does this derive from a depletion curve? Please, add a reference.
The equation is directly from ERA5 documentation. We will revise the text and add a reference.

L159 very close: please quantify!

We will quantify the difference and edit the text.

Table 1. I would add the period of availability of the different data sources.

We will edit the table according to the comment.

L190 I am wondering if it might be problematic to use products that assimilate another reanalysis product (ERA-Interim) as "reference". Might be that the differences that you obtain are due to the assimilation of the ERA-Interim product? I think this point needs further discussion.

We will add more discussion on this issue.

L212 How can you explain that the differences are negligible?

We will quantify the difference and edit the text.

L237 What is the reason why you choose the nearest neighbor interpolation instead of a cubic for example?

The nearest neighbor was sufficient for our study because the native grid resolutions of the datasets were already very similar. The higher-resolution MODIS albedo product was first coarsened to 0.25° resolution by calculating the mean value of all the grid cells within one 0.25° × 0.25° grid cell and subsequently resampled to 25 km equal-area projection using the nearest neighbor method.

I am wondering also why you haven't used the snow CCI SCF product as additional reference dataset.

We have only used two reference datasets to keep the number of datasets reasonable.

**Section 3 Results**

I would appreciate to see some more metrics (RMSE, correlation).

We will add metrics as suggested.

L308 Please, explain better what you mean with "explained with the uncertainties".

We will edit the text so that the meaning will be clear for the readers.

**Discussion**

L488 quantify "typically" or add a reference

The reference (Key et al., 2001) is after the next sentence. We will reword these sentences.

---

## Author Response (AR2)

The review comments are shown in black and the author responses in red.

We would like to thank the reviewer for reviewing our manuscript. Please find below our responses to the comments.

This revision has improved acknowledgement of limitations in the study and I recommend that it is now largely acceptable. I have a few more minor comments.

Lines 176 ("SnowCCI … provided in a geographical latitude-longitude grid") and 187 ("SnowCCI data … are mapped to a 25 km EASE-Grid") appear contradictory (I think because the distinction between SnowCCI and bias-corrected SnowCCI is not very clear).
We edited the text according to the comment.

Line 209
Indeed, "the longest running satellite-based record of any environmental variable" (Estilow et al. 2015).
We edited the text according to the comment.

Line 300
If referring to a Supplementary figure in this much detail, it shouldn't be supplementary. Is it surprising that assimilation of IMS snow cover improves SWE but not SCE in this figure?
We moved Figure S3 to the main manuscript and edited the text.

Colour in figures should mean something. Repeatedly changing the colours that represent ERA5 and ERA5-Land within Figures 2-7, 10, 11, S1 and S3 forces the reader to keep recalibrating.
We changed the colors of the timeseries figures so that the colors are the same in every figure.

Colour bars in Figures 1 and S2 should have labels (SWE, SCE, albedo).
We added labels.

The caption of Figure S1 is very uninformative.
We changed it to: "Monthly timeseries of different albedo estimates in ERA5 and ERA5-Land."